# Flexible calorimetric flow sensor with unprecedented sensitivity and directional resolution for multiple flight parameter detection

Zheng Gong [1], Weicheng Di[2], Yonggang Jiang [1,3] ✉, Zihao Dong[1], Zhen Yang [1,4], Hong Ye[1], Hengrui Zhang[1], Haoji Liu[2], Zixing Wei[2], Zhan Tu [5], Daochun Li[2], Jinwu Xiang[2], Xilun Ding [1], Deyuan Zhang[1] & Huawei Chen [1]

The accurate perception of multiple flight parameters, such as the angle of attack, angle of sideslip, and airflow velocity, is essential for the flight control of micro air vehicles, which conventionally rely on arrays of pressure or airflow velocity sensors. Here, we present the estimation of multiple flight parameters using a single flexible calorimetric flow sensor featuring a sophisticated structural design with a suspended array of highly sensitive vanadium oxide thermistors. The proposed sensor achieves an unprecedented velocity resolution of 0.11 mm·s⁻¹ and angular resolution of 0.1°. By attaching the sensor to a wing model, the angles of attack and slip were estimated simultaneously. The triaxial flight velocities and wing vibrations can also be estimated by sensing the relative airflow velocity due to its high sensitivity and fast response. Overall, the proposed sensor has many promising applications in weak airflow sensing and flight control of micro air vehicles.

The development of micro air vehicles (MAVs) that can sense and manipulate airflow in a manner similar to birds has long been a goal of aeronautical researchers. In contrast to conventional high-speed airplanes, MAVs significantly lower airspeeds are more susceptible to environmental turbulence[1]. Although significant progress has been made in the fields of vision-based flight control and the miniaturisation of inertial measurement units (IMUs), highly sensitive and miniaturised flow sensors for airflow sensing are urgently desired to provide comprehensive aerodynamic information. Mimicking the function of insect mechanoreceptors (Johnston's organ) for sensing airflow stimuli, quadrotor drones with integrated pressure sensors have demonstrated creature-like obstacle avoidance capabilities by virtue of flow field perception[2]. Additionally, the estimation of multiple flight parameters using an array of pressure and flow sensors has been proposed as a feasible solution for intelligent flight control[3,4]. However, the

pressure stimuli are very small at low airspeeds and realising high-sensitivity flexible pressure sensors using flexible electronics is challenging.

Inspired by the flow perception mechanisms of birds, insects, and bats, distributed and highly sensitive airflow velocity sensors have been used to cope with complex flow fields. Artificial hair flow sensors have been studied extensively to mimic the fluid–structure interaction principle of biological flow-sensing hairs[3,5–9]. However, the sensitivity of a hair flow sensor is proportional to the square of the flow velocity, leading to low sensitivity at low flow velocities[3]. Flexible hot-film sensors have also been investigated and introduced into MAV applications[10,11]. However, because such sensors are unable to measure the flow direction, they are typically arranged in complex arrays for flight parameter estimation, which impedes their practical application. Additionally, to achieve multimodal sensing functions

[1]School of Mechanical Engineering and Automation, Beihang University, Beijing 100191, China. [2]School of Aeronautic Science and Engineering, Beihang University, Beijing 100191, China. [3]International Research Institute for Multidisciplinary Science, Beihang University, Beijing 100191, China. [4]Zhiyuan Research Institute, Hangzhou 310013, China. [5]Institute of Unmanned Systems, Beihang University, Beijing 100191, China. ✉e-mail: jiangyg@buaa.edu.cn

such as the relative airflow velocity, angle of attack (AOA), angle of sideslip (AOS), and wing vibration, the integration of different physical sensors such as pressure, hot-film, and piezoelectric sensors is required to develop smart MAV skin[12–15]. We previously reported a sensing fusion methodology for AOA and airspeed using an array of flow and pressure sensors[4,5]. However, the detection of multiple flight parameters using a single flow sensor, which can significantly simplify air data sensing systems for lightweight MAVs, has never been reported.

To satisfy the multi-parameter perception requirements of MAVs, we propose a thin flexible calorimetric flow (FCF) sensor with high sensitivity and directionality. The proposed sensor has a structure with a suspended array of highly sensitive vanadium oxide ($VO_x$)-based thermistors on a thin flexible substrate. Benefiting from the high-temperature coefficient of resistance (TCR) of the sputtered $VO_x$ film, the FCF sensor exhibits high flow velocity and directionality resolutions (0.11 mm·s⁻¹ and 0.1°, respectively). By attaching a single FCF sensor to the leading edge of the airfoil, the AOA and AOS can be computed simultaneously. With two FCF sensors mounted on an MAV wing, the MAV can precisely estimate the relative airflow velocity (mean error <0.2 m·s⁻¹), achieving the highest accuracy among state-of-the-art airflow velocity measurement methods (Supplementary Table 1)[10,12,15–17]. Furthermore, with its high resolution and fast response characteristics, the FCF sensor can measure weak high-frequency airflow, providing an valuable sensing function for the vibration of MAV wings.

## Results

### Flow sensor design

To realise the high-precision measurement of two-dimensional flow velocity, we designed the FCF sensor as a multilayered structure with arrayed highly sensitive thermistors on a thin polyimide (PI) substrate (Fig. 1a). The sensor consists of a spiral heater, an array of $VO_x$ thermistors, flexible PI supporting and protection layers, and a copper-on-PI (COP) flexible substrate. We isolated the heater and thermal sensor from a flexible substrate using a channel to mitigate thermal conduction via the PI substrate. Additionally, the TCR of thermistors is a significant factor affecting the sensitivity of the FCF sensor. Many semiconductor oxide materials have TCRs several times higher than those of metals (Supplementary Table 2)[18–24]. However, they typically have large resistivity, which results in high 1/f noise and impedance mismatch[18,25]. $VO_x$ combines a high TCR (−2% K⁻¹) with low 1/f noise, providing significant advantages for temperature sensing applications[26,27], and was therefore employed as the material for integrated thermistors on a flexible substrate. Because six pairs of $VO_x$ thermistors are symmetrically cross-arranged around the heater, the temperature difference $\Delta T$ between the upstream and downstream thermistors can accurately reflect heat convection while cancelling out ambient thermal influence. The directionality of flow can be inferred from the difference in $\Delta T$ between cross-arranged thermistor pairs (Fig. 1b). The centre-to-edge distances $D$ from the $VO_x$ thermistors to the central heater are 120, 320, and 520 μm (Fig. 1c). As one of the key design parameters, the centre-to-edge distance $D$ significantly affects

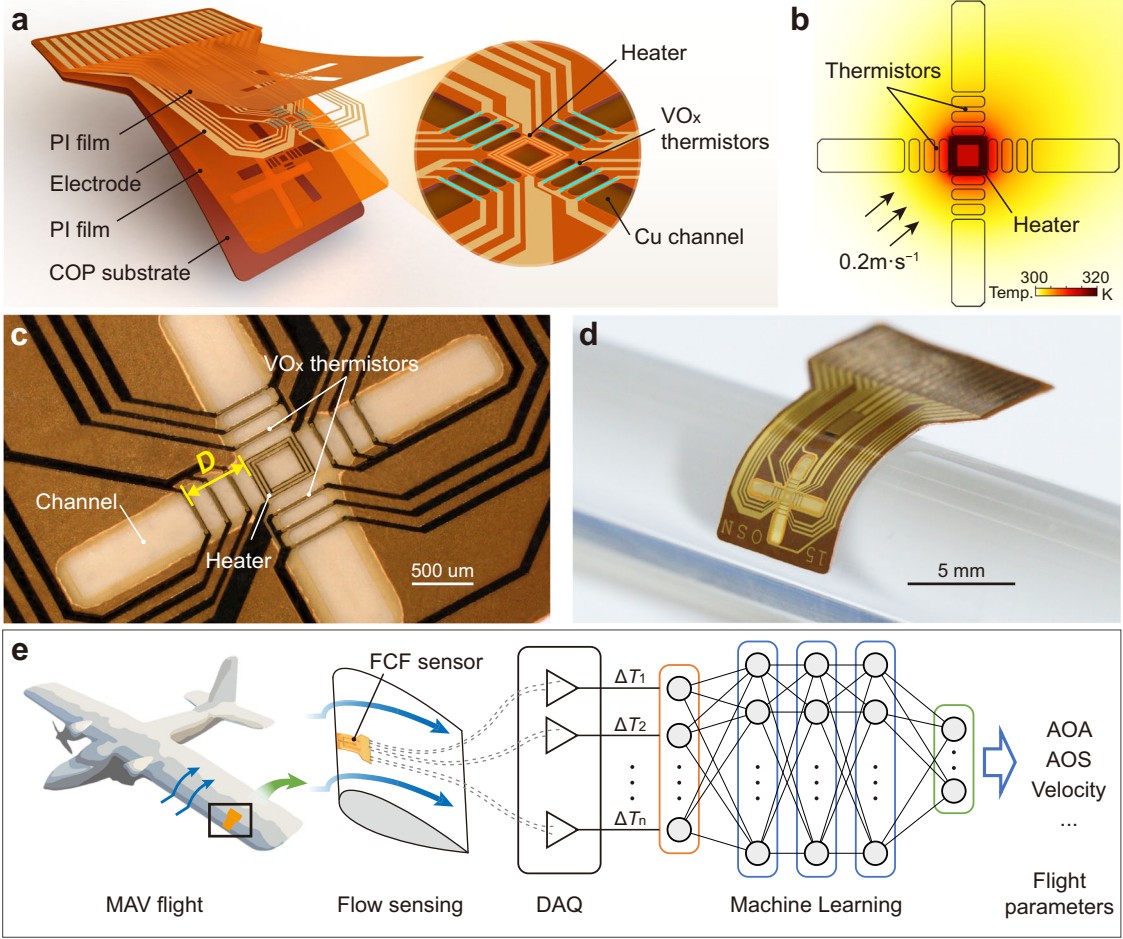

**Fig. 1 | Flexible calorimetric flow sensor design. a** Exploded view of the FCF sensor. **b** Illustration of the working principle of the FCF sensor. Spatial map of the temperature at a velocity of 0.2 m·s⁻¹. **c** Optical image of the heater, channels, and $VO_x$ thermistors. *D* represents the centre-to-edge distances from the $VO_x$ thermistors to the central heater. **d** FCF sensor attached to a glass cylinder with a radius of 6 mm. **e** Conceptual diagram of multiple flight-parameter estimation using a single FCF sensor. $\Delta T$ represents temperature difference between the upstream and downstream thermistors.

the sensitivity and directionality characteristics of calorimetric flow sensors (see Supplementary Fig. 1 and Note. 1 for details regarding the heat transfer model)[28–31]. This design enables the sensor to obtain nonlinearly correlated multi-channel data, which can enhance the estimation accuracy of flight parameters.

We propose a facile fabrication process on a flexible COP substrate facilitated by the low-temperature (<300 °C) preparation of $VO_x$ thin-film thermistors and a well-defined suspended structure (see Supplementary Fig. 2 for details regarding $VO_x$ thin-film preparation and characterisation). The copper in COP is used as a sacrificial layer, which ensures that the total thickness of the supporting and protection PI layers is within 10 μm in addition to the well-defined thickness of the thermal isolation channel (see Supplementary Fig. 3 for the fabrication details). Because the total thickness of the FCF sensor is less than 90 μm, it is sufficiently flexible to be attached to curved surfaces (Fig. 1d). As shown in Fig. 1e, the FCF sensor is conformally attached to the leading edge of an MAV wing. Multiple flight parameters such as AOA, AOS, and relative airflow can be estimated using a pre-trained machine-learning-based neural network model.

## Sensing characteristics of the FCF sensor

For characterisation in a benchtop wind tunnel (Fig. 2a), the FCF sensor was attached to the leading edge of a sharp plate to reduce the effects of the aerodynamic boundary layer (Supplementary Fig. 4C). A feedback circuit was used to enable the FCF sensor to operate in the constant-temperature-difference (CTD) mode (Supplementary Fig. 5A)[32,33]. The three pairs of thermistors in order from near to far from the heater were defined as P1, P2, and P3 and were connected to differential amplifier circuits (Supplementary Fig. 5B). Figure 2b indicates that the output voltages of the three thermistor pairs for airflow velocities range from 0 to 30 m·s⁻¹, indicating that the highest

sensitivity is generally obtained from the thermistor pair nearest to the heater. Detailed measurements in a small range of airflow velocities are presented in Fig. 2c, demonstrating that the differential output of P1 can be monitored as a linear function in the flow velocity range of 0–1.25 m·s⁻¹ with a sensitivity of 1.817 V·m⁻¹ s. The minimum detectable velocity was calculated to be 0.11 mm·s⁻¹ based on the noise level of the P1 ($9.8 \times 10^{-5}$ V) (Supplementary Fig. 6A) and $2\sigma$ criterion[34]. However, the differential output of P2 ($D = 320$ μm) is higher than that of P1 ($D = 120$ μm) when the flow velocity is less than 0.2 m·s⁻¹ (see the inset in Fig. 2d), which is supported by our design model (Supplementary Fig. 1). Both the calculation and simulation results demonstrate that the optimal distance $D$ between the heater and thermistor pair with the maximum temperature difference decreases as the flow velocity increases (Fig. 2d).

The FCF sensor exhibited a highly repeatable response when subjected to cyclic flow velocities of 1, 5, and 10 m·s⁻¹ (Fig. 2e). The repeatability standard deviations were approximately 0.5% of the measured flow velocity values. In addition to the merits of sensitivity and repeatability, the response time is another important parameter for FCF sensors. The response time of a calorimetric flow sensor can be defined as the time constant $T$ (or $T'$) required for the output voltage to increase to 63.2% (or decrease to 36.8%) of its initial amplitude[35]. We measured the response time in the pulse-operated CTD mode with the heating system on and off at a constant incoming flow velocity (Fig. 2f). The $VO_x$ thermistor pair P1 closest to the heater exhibited short-time constants with $T_1$ and $T_1'$ values of 16 and 22 ms, respectively. Although the time constants increase with increasing distance $D$, the maximum time constant is still less than 37 ms for thermistor pair P3. Supplementary Fig. 6B shows the heating power of the FCF sensor in the range of 3–4.4 mW from 0 to 30 m·s⁻¹. This low power consumption is also essential for energy conservation and flight endurance of the MAV.

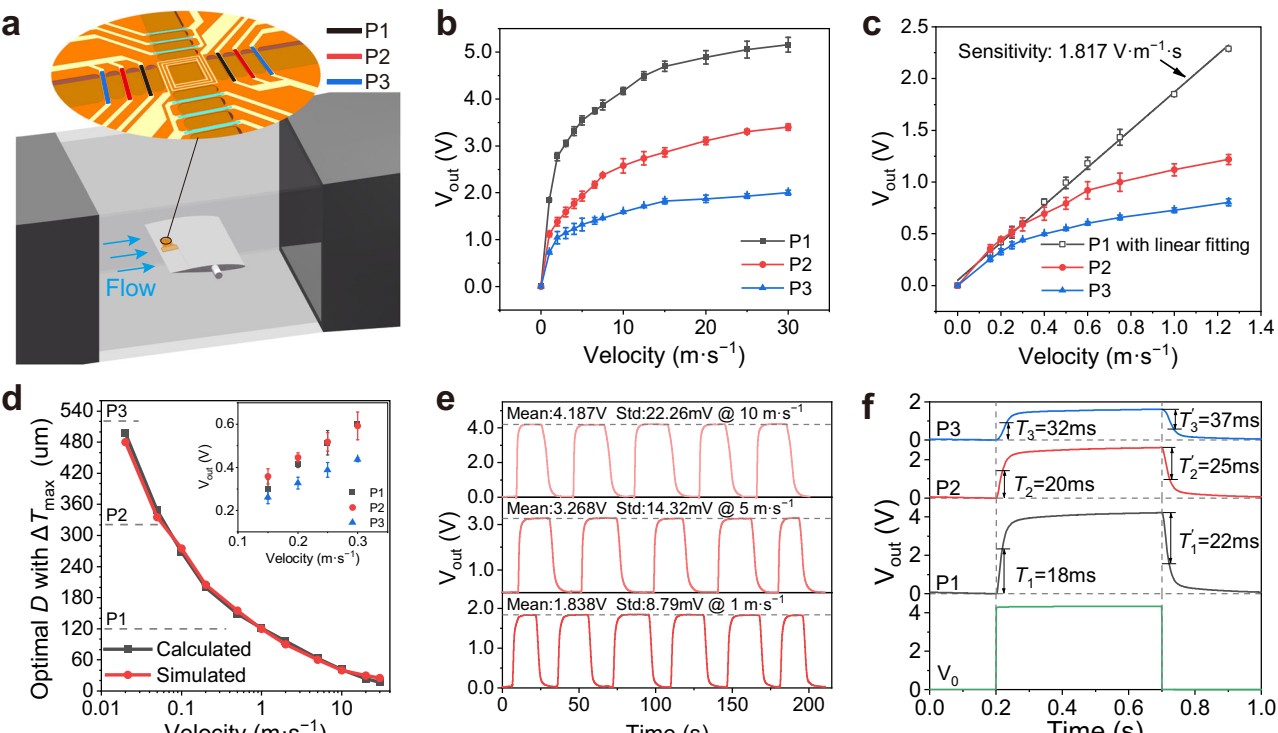

**Fig. 2 | FCF sensor characterisation and performance. a** Schematic of the experimental setup for characterisation of the FCF sensor. **b** Comparison of the output voltages $V_{out}$ of three $VO_x$ thermistor pairs (P1–3) within the flow velocity range of 0–30 m·s⁻¹ in CTD mode. **c** Output voltages within the flow velocity range of 0–1.25 m·s⁻¹, where the output of P1 is linearly related to the flow velocity. **d** Simulation and calculation of the optimal distance $D$ between the heater and

thermistor pair with maximum output $\Delta T_{max}$ at different flow velocities, where the inset shows the output voltages of thermistor pairs measured at low flow velocity. Error bars in (**b**, **c**, **d**) represent standard deviation, $n = 3$ independent replicates. **e** Output voltages of P1 for cyclic loading at different flow velocities of 1, 5, and 10 m·s⁻¹. **f** Dynamic response of the CTD circuit $V_0$ and three thermistor pairs (P1–3) when pulsed-switching the heater at a flow velocity of 10 m·s⁻¹.

Supplementary Table 3 presents comprehensive comparisons of the sensing modalities, structures, materials, maximum sensitivities, minimum detectable velocities, repeatability errors, heating power, maximum angle errors, and detection ranges of reported calorimetric flow sensors[34,36–41]. Our FCF sensor provides unprecedented sensitivity and repeatability as a result of its suspended structure and $VO_x$ thermistor material, which are essential for high-precision airflow perception.

## Directionality of the FCF sensor

Changes in the flow direction affect the temperature distribution on the surface of the FCF sensor, and the velocity components of orthogonal decomposition can be measured by thermistor pairs perpendicular to each other. The directionality performance of the FCF sensor was characterised by mounting it in the middle of a slightly raised disk surface (Fig. 3a). Changes in the airflow angle $\theta$ were achieved by rotating the FCF sensor (Fig. 3b and Supplementary Fig. 4). As shown in Fig. 3c, the output voltage of the thermistor pair P1–3 (or V1–3) has a simple cosine (or sine) dependence on the airflow angle $\theta$ at a low flow velocity of 0.2 m·s⁻¹. The experimental results agree well with computational fluid dynamics (CFD) simulation results in terms of the temperature difference (Fig. 3d). Under a low flow velocity, thermistor pairs P3 and V3, which are the farthest from the heater, have the highest angular sensitivities, where the maximum angular sensitivity of V3 is 3.4 mV·deg⁻¹. Based on the measured noise levels (Supplementary Fig. 6C), the directional resolution at 0.2 m·s⁻¹ was calculated to be 0.79°. The output voltages of the thermistor pairs increasing the incoming flow velocity to 10 m·s⁻¹ are presented in Fig. 3e. One can see

that P1 and V1, which are closest to the heater, are the most sensitive, where the maximum angular sensitivity of V1 is 36.7 mV·deg⁻¹. The directional resolution at 10 m·s⁻¹ was calculated to be 0.1° based on the noise level presented in Supplementary Fig. 6B. Furthermore, we evaluated the angular measurement accuracy of the FCF sensor, and the maximum angular error was less than 1.6° (Supplementary Fig. 6C). When the flow velocity increased to 10 m·s⁻¹, the thermistor pairs at different distance $D$ have different direction response functions. The output voltages of P1 (or V1) still exhibit cosine (or sine) dependence on the airflow angle $\theta$, whereas the output voltages of the thermistor pairs far from the heater exhibit cubed-cosine (or cubed-sine) dependence on the airflow angle $\theta$ (Fig. 3d, e). The simulated thermistor temperature profiles clearly explain this phenomenon (Fig. 3f). High-velocity airflow restricts heat diffusion to a narrow region downstream of the heater. When the airflow angle is in the range of 50°–130° (or 230°–310°), the thermistor pairs far from the heater (V2 and V3) are inside this narrow thermal diffusion zone and the temperature difference varies significantly with the angle. When the airflow angle is in the range of 130°–230°, these thermistors are outside the thermal diffusion zone and the temperature difference changes very little with the angle. However, because V1 is the closest to the heater and can contact the thermal diffusion zone over almost the entire angular range, the temperature difference curve maintains a sinusoidal profile, which is consistent with the experimental results.

## Estimation of AOA and AOS

The high sensitivity and directional resolution of the FCF sensor make it particularly suitable for detecting multiple flight parameters. As

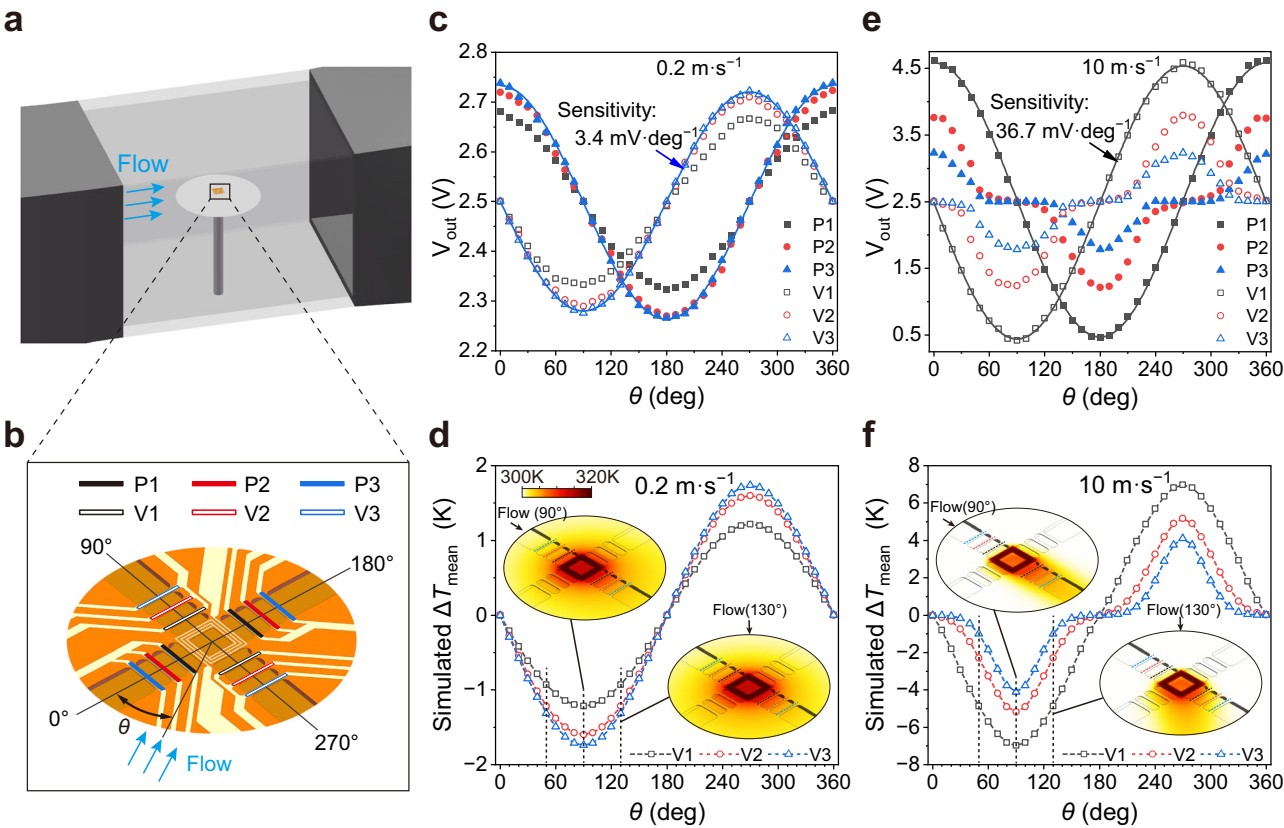

**Fig. 3 | Directionality of the FCF sensor. a** Schematic of the experimental setup for directionality characterisation of the FCF sensor. **b** Regional enlarged view of FCF sensor, where the three thermistor pairs along the 0° airflow angle are P1–3 and those along the 90° airflow angle are V1–3. **c** Variation of output voltage $V_{out}$ with airflow angle $\theta$ at 0.2 m·s⁻¹. **d** Simulated mean temperature difference $\Delta T_{mean}$ curves at the positions of the V1–3 thermistor pairs at 0.2 m·s⁻¹. The colour maps

correspond to the local temperature distribution on the surface of the FCF sensor at airflow angles of 90° and 130°. **e** Variation of output voltage with airflow angle $\theta$ at 10 m·s⁻¹. **f** Simulated $\Delta T_{mean}$ curves at the positions of the V1–3 thermistor pairs at 10 m·s⁻¹, where 50°–130° and 230°–310° are the highly sensitive response intervals of V2 and V3.

shown in Fig. 4a, the FCF sensor was conformally attached to the leading edge of a wing model (Supplementary Fig. 4E). The FCF sensor for two-dimensional flow sensing has three pairs of thermistors in each direction, among which thermistor pairs P1–3 are sensitive to AOA variations, whereas V1–3 are sensitive to the AOS variations. Because a multilayer perception (MLP) neural network can achieve highly non-linear mapping from inputs to outputs and has strong generalisation capabilities[42], we employed an MLP to estimate the AOA and AOS simultaneously at different airflow velocities. The AOA and AOS were varied in the range of ±20° with a step of 2°, and a total of 441 datasets (21 × 21 grey points) were obtained as a training set (Fig. 4b). We then adjusted the AOA and AOS to obtain data for 184 testing points (blue points). The signals of the six thermistor pairs (P1–3 and V1–3) were normalised and then input into the MLP model to estimate the AOA and AOS simultaneously. The estimates of the testing points at a flow velocity of 10 m·s⁻¹ are presented in Fig. 4c and agree well with the actual values. To verify the effect of the number of thermistor pairs on the prediction accuracy, we compared the estimation results when using the signals of thermistor pairs P1 and V1 (PV1), P2 and V2 (PV2), P3 and V3 (PV3), and all thermistor pairs (PV1–3) as inputs. The mean absolute errors (MAEs) of the AOA and AOS are 0.54° and 0.56°, respectively, when using PV1–3 as inputs, which are much smaller than those when using PV1, PV2, or PV3 as inputs (Fig. 4d), suggesting that the array design of the thermistor pairs in a single FCF sensor provides promising guidance for improving the accuracy of AOA and AOS recognition. When the airflow velocity is increased to 30 m·s⁻¹, the estimated and actual values of the AOA and AOS remain well matched

(Fig. 4e). As shown in Fig. 4f, the estimation when using the signals of all thermistor pairs exhibits the best accuracy, where the MAEs of the AOA and AOS are 0.58° and 0.53°, respectively. These results demonstrate that the FCF sensor has good stability and can maintain high estimation accuracy for the AOA and AOS at different flow velocities. Furthermore, the flight parameters were resolved using only one FCF sensor, which is superior to existing techniques that require arrays of flow sensors (Supplementary Table 1).

## Relative airflow velocity estimation and wing vibration monitoring

Relative airflow velocity is an essential parameter for the agile flight control of MAVs. In this section, we describe the accurate detection of tri-axis relative airflow velocity using the highly sensitive FCF sensor. As shown in Fig. 5a, two FCF sensors were mounted on a fly-wing MAV with dual vector rotors. Sensor #1 measured the velocity component in the x-y direction, and sensor #2 measured the velocity component in the y-z direction. To avoid interference from the propeller airflow during flight, both FCF sensors were arranged at the edge of the wing and an external IMU module was installed next to sensor #2 to collect the vibration information of the wing simultaneously. Heating control, signal collection, and data storage were achieved using an onboard electronic system (Supplementary Fig. 7). The two FCF sensors were calibrated by fixing the airframe and moving it in fixed directions on a servo slide rail (Supplementary Fig. 8). The reciprocating movement velocities of the airframe along the three axes of the body coordinate system and corresponding FCF sensor outputs are presented in Fig. 5b.

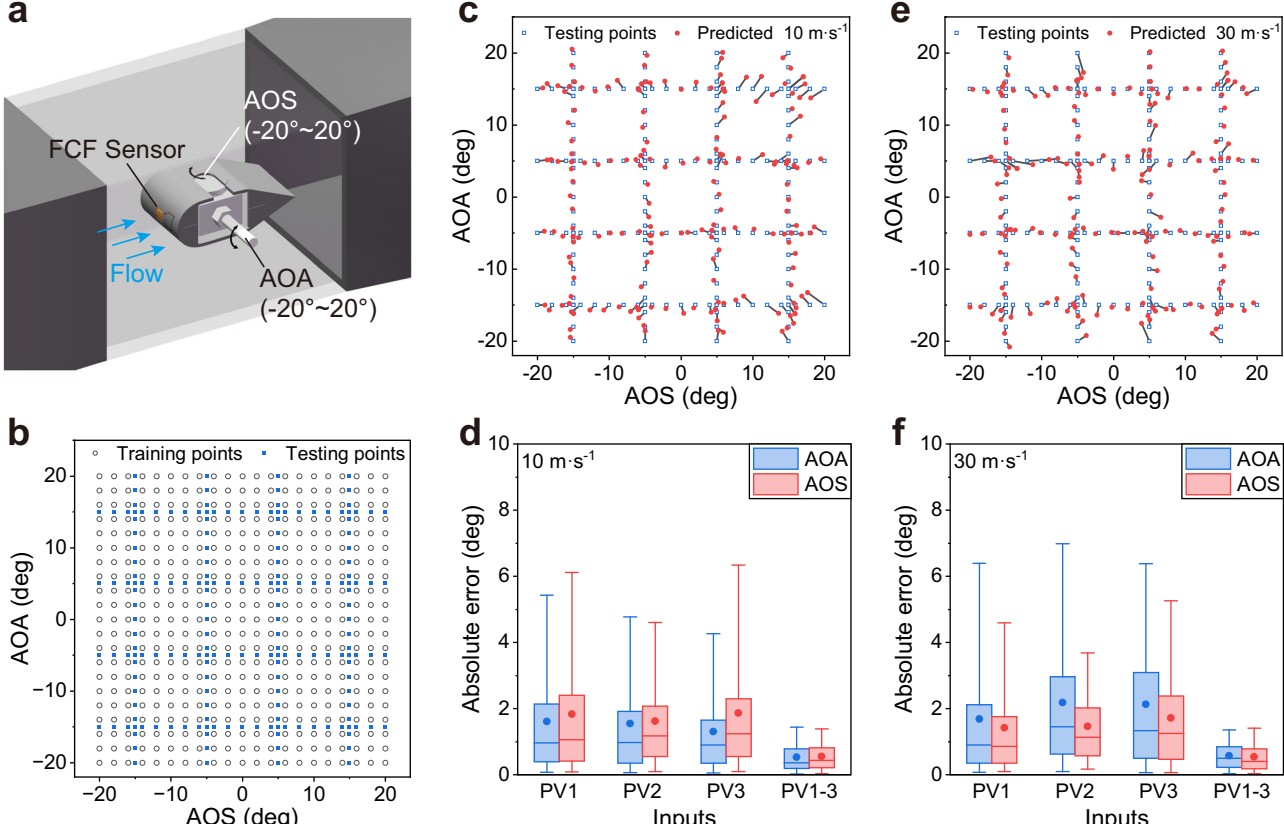

**Fig. 4 | Estimation of AOA and AOS using a single FCF sensor. a** Schematic of the experimental setup for AOA and AOS predictions, where the FCF sensor is attached to a NACA 0016-MOD airfoil with a radius of 14 mm. **b** Training (grey) and testing (blue) points for different AOA and AOS from −20° to +20°. **c** Distribution of estimated results (red points) using signals of PV1–3 as inputs at 10 m·s⁻¹. **d** Comparison of the absolute errors of AOA and AOS of testing points with different inputs at

10 m·s⁻¹. **e** Distribution of estimated results using signals of PV1–3 as inputs at 30 m·s⁻¹. **f** Comparison of the absolute errors of AOA and AOS of testing points with different inputs at 30 m·s⁻¹. Box plots indicate median (middle line), 25th, 75th percentile (box) and 5th and 95th percentile (whiskers) as well as mean (points). Number of samples $n = 184$.

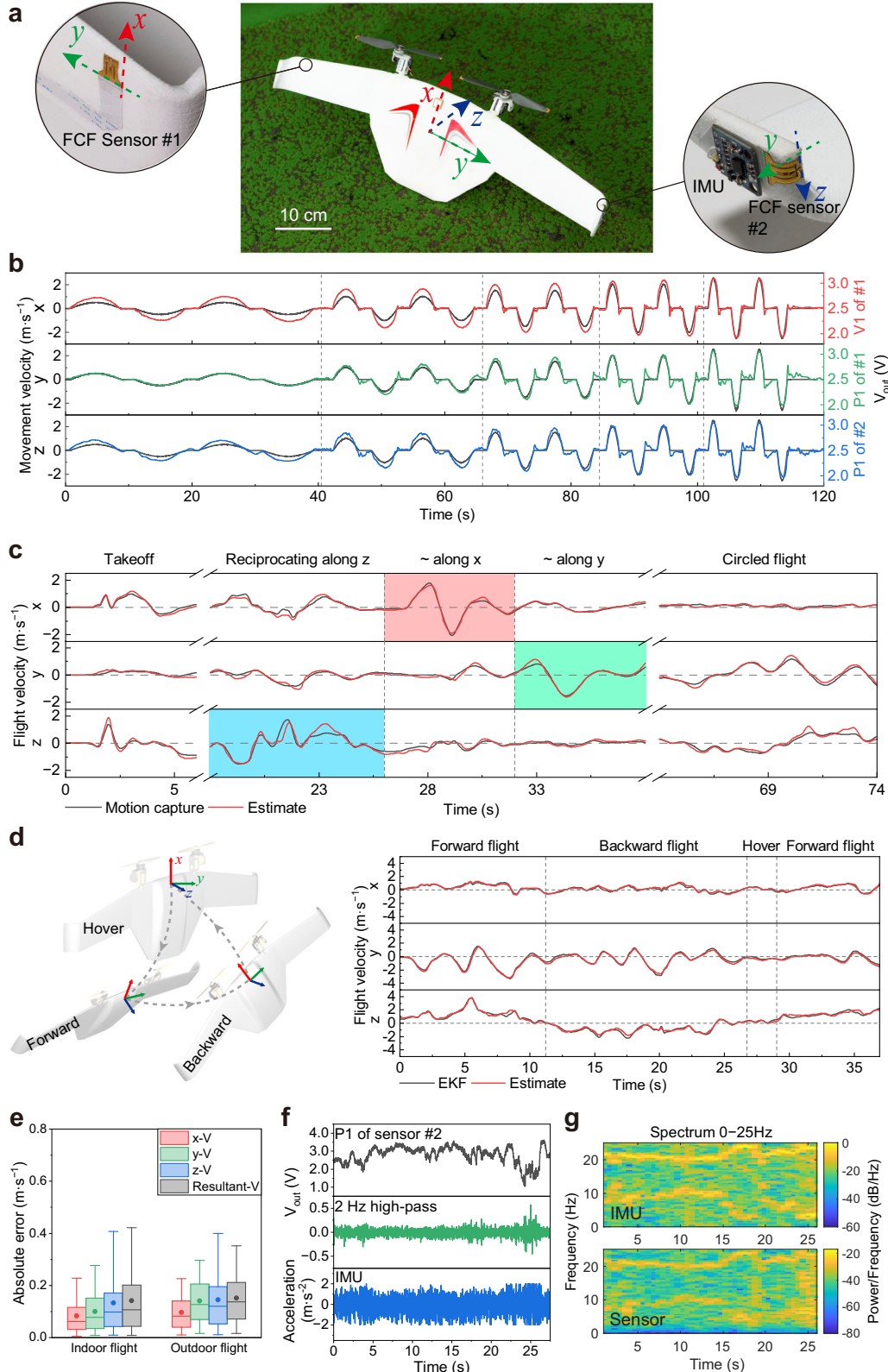

The two sensor components exhibit different nonlinear responses to the relative airflow velocities in three directions. However, no significant phase lag exists between the motion velocity and output voltage based on the fast response of the FCF sensor (Fig. 2f).

To demonstrate the airflow velocity estimation capabilities of the FCF sensor, we controlled the MAV with a joystick to perform takeoff, reciprocating flight and circled flight operations sequentially

(Supplementary Movie 1). Because this experiment was conducted indoors, the relative airflow velocity was assumed to be opposite to the motion velocity. Therefore, we compared the flight velocity of the MAV obtained from a motion capture system to the relative airflow velocities estimated using an MLP neural-network-based strategy. Figure 5c presents the velocity components of the MAV along the three axes in the body coordinate system. The estimated values

**Fig. 5 | Relative airflow velocity estimation and wing vibration perception.**
**a** MAV equipped with two FCF sensors and an external IMU module. **b** Comparison of the movement velocity with the output voltages of the FCF sensors when the MAV is fixed on the mount for tri-axis movement. **c** Comparison of the flight velocity calculated by the motion capture system and relative velocity estimated by the FCF sensors when the MAV performs takeoff, reciprocating motion along the z (blue), x (red), and y (green) axes, and circled fight indoors. **d** Comparison of the flight velocity calculated by the EKF method with the relative velocity estimated by the FCF sensor when the MAV performs forward flight, backward flight, and hovering outdoors. **e** Absolute errors of the indoor and outdoor flight velocities

between the calculated and estimated values. Box plots indicate median (middle line), 25th, 75th percentile (box) and 5th and 95th percentile (whiskers) as well as mean (points). Number of samples n = 1803 for indoor flight and n = 1851 for outdoor flight. **f** Output voltage of P1 (sensitive to z-axis velocity component) of FCF sensor #2, the 2 Hz high-pass filtered signal, and the acceleration signal acquired by the IMU synchronously during MAV flight. **g** Comparison of the spectra of signals from IMU and P1 of FCF sensor #2, illustrating the wing vibration with 10 and 20 Hz as the characteristic frequencies. IMU inertial measurement unit, EKF extended Kalman filter.

exhibit excellent consistency with the relative flight velocities for different flight motions. This finding indicates that the FCF sensor can accurately identify the relative airflow experienced by the MAV during flight.

To demonstrate the ability of our sensor to estimate relative airflow velocities further, we conducted outdoor experiments using the same MAV model and sensor configuration at higher flight velocities under windless conditions (Supplementary Movie 2). We manually guided the MAV to perform a fast-reciprocating flight manoeuvre along the z-axis of the body coordinate system, as well as a short hover and a forward flight. Figure 5d presents the results of the outdoor flight experiments. The black lines represent the flight velocity components of the body coordinate system obtained using an extended Kalman filter (EKF) algorithm that fuses measurements from the onboard Global navigation satellite system (GNSS), IMU, and manometer. The red lines represent the relative airflow velocities estimated by the MLP neural network, which agree well with the flight velocities calculated using the EKF. The z-axis velocity exhibits a peak value of 3.71 m·s$^{-1}$ at 5.4 s with only a 0.5% difference between the estimated and calculated values. We calculated the MAEs of the estimated velocities based on the indoor flight velocities captured by the motion capture system and outdoor flight velocities computed using the EKF (Fig. 5e). The statistical results reveal that the MAEs of the velocity components of each axis and resultant velocity for indoor flight are lower than 0.15 m·s$^{-1}$ and that the MAEs for outdoor flight are lower than 0.2 m·s$^{-1}$, which are superior to the results of previous studies (Supplementary Table 1). The outdoor experiment results further demonstrate the excellent reliability and stability of our FCF sensor for airflow sensing and flight parameter estimation.

We further evaluated flight path drift by integrating the estimated velocities (Supplementary Fig. 9). Since these estimates are given in body coordinate frame, they were first transformed in inertial frame using the rotation matrix, and then numerically integrated. In the indoor flight phase, the mean drifts of the path obtained by flight velocity with respect to the ground truth were less than 0.07 m·s$^{-1}$. During outdoor flight, the mean drifts of the path based on velocity integration to the path given by EKF and GNSS were calculated to be 0.1161 and 0.4872 m·s$^{-1}$, respectively. The MLP network for flight velocity estimation is pre-trained with the velocity given by the EKF, therefore the path estimation based on flight velocity is closer to the path given by the EKF. However, it is difficult for us to assess the drift between the estimated path and the ground truth for outdoor flight due to the unavoidable drift of consumer-grade GNSS, IMUs, etc.[43]. Moreover, the indoor path estimation results reflect the feasibility of using the obtained flight velocity estimates for odometry.

In addition to estimating relative airflow velocity during flight, the fast response and high sensitivity of the proposed FCF sensor facilitate the analysis of wing vibration information by sensing relative airflow movement. When the MAV was hovering, the velocity signal of sensor #2 along the z-axis of the body coordinate system was collected and processed using 2 Hz high-pass filtering, as shown in Fig. 5f. The short-time Fourier transform of the filtered sensor signal was calculated to obtain the spectrogram shown in Fig. 5g. Obvious intensity peaks exist at 10 and 20 Hz, indicating that the wing was vibrating at the

corresponding frequencies. For comparison, we plotted the spectrum of the acceleration signal acquired simultaneously by the commercial IMU module and found that the peak frequencies of the acceleration signal were consistent with those of the FCF sensor signal. These results indicate that the FCF sensor has such a fast response and unprecedented resolution that it can detect both relative flight velocity and wing vibration.

## Discussion

We developed a thin FCF sensor with a well-designed $VO_x$ thermistor array for flight parameter estimation. The proposed FCF sensor achieved a high-velocity resolution (0.11 mm·s$^{-1}$), small repeatability standard deviation (0.5%), fast response time (20 ms), and high directional resolution (0.1°). These excellent characteristics facilitate the practical application of FCF sensors for the estimation of MAV flight parameters. The FCF sensor was attached to a curved airfoil to perform typical flight parameter perception functions, including AOA, AOS, and relative airflow velocity sensing, based on machine-learning algorithms. Interestingly, the proposed FCF sensor can also identify the vibration information of an airfoil.

This excellent sensing performance was attributed to the $VO_x$ thermistors with a high TCR and suspended structure with effective thermal isolation. The processing of $VO_x$ thin films has traditionally involved high temperatures of over 400 °C, which are incompatible with the processing of sensors on flexible PI substrates. Therefore, we developed low-temperature sputtering and annealing processes to solve this problem and achieved a high TCR of 1.9% K$^{-1}$. Additionally, we conducted structural optimisation to enhance the temperature differences between thermistor pairs to improve the sensitivity of the FCF sensor. The sensitivity of calorimetric flow sensors is typically increased by introducing trenches or insulating cavities to reduce heat dissipation from the substrate[30,36]. Our proposed suspended structure not only mitigates thermal conduction from the PI substrate but also enhances convective heat transfer through both the upper and lower sides of the thermistor, thereby further increasing the temperature difference of the thermistor pairs. To verify this effect, we established a general 1D heat transfer model of an FCF sensor with a suspended structure, which was highly consistent with the CFD simulation model in terms of the key design parameters. We determined that the PI-compatible processing of the $VO_x$ thermistor and suspended structure design significantly enhanced the sensing performance of the FCF sensor.

In addition to high-sensitivity characteristics, an array design of sensing units on a thin flexible PI substrate provides a novel strategy for high-precision MAV flight parameter estimation using a single FCF sensor. We arranged six pairs of thermistors on one FCF sensor with different sensitivities and directional flow-sensing characteristics. This array design not only facilitates the regulation of sensing performance by directly adjusting the distance between thermistor pairs and the heater but also provides rich information and high-accuracy prediction of flight parameters based on neural network algorithms.

For MAV flight parameter detections, the system can collect and store multi-channel data by integrating onboard sampling circuits and a microcontroller. In future work, we will port the artificial neural

network model to the microcontroller to achieve online flight velocity and attitude estimation, taking a step towards practicality.

In summary, this paper reported a thin, flexible, and calorimetric flow sensor with high sensitivity and directionality. An FCF sensor can perform multi-parameter sensing tasks for MAVs with high accuracy, including attitude angle estimation, relative airflow velocity estimation, and wing vibration monitoring. This study provides a promising strategy for flow sensor design in the field of airflow sensing for MAVs, thereby broadening the applications of FCF sensors to attitude detection, airspeed estimation, and safety monitoring.

## Methods

### Preparation of VO$_x$ films
VO$_x$ films were deposited on PI film substrates via reactive ion beam sputtering using a V target (99.9%) in an O$_2$/Ar gas mixture. The sputtering pressure was set to 0.035 Pa. The partial pressure of oxygen (PaO$_2$) was regulated using a mass flow meter. After deposition, the films were annealed in an Ar atmosphere at a set temperature for 2 h.

### Fabrication of FCF sensors
High-temperature-resistant epoxy (540, KAISIMI) was spin-coated onto a glass substrate at 3000 rpm for 30 s. Next, the COP layer was uniformly laminated on the glass substrate via roll pressing and cured at 90 °C for 4 h. The PI precursor solution was spin-coated at 3000 rpm for 30 s on the COP layer and then prebaked at 150 °C on a hot plate for 10 min to remove the organic solvent, after which the PI film was fully cured at 280 °C on a hot plate under ambient conditions for 30 min to form the supporting layer. A lift-off process was employed to form electrodes and resistive heater metal layers (Cr/Au; 20 nm/300 nm), and VO$_x$ thin-film (400 nm) thermistors were also patterned using a lift-off process. A second layer of PI, which was used as the encapsulated layer, was then formed via spin-coating. Both the supporting and encapsulated PI layers were patterned using reactive ion etching in O$_2$/SF$_6$ gas at ambient temperature for 50 min. The exposed Cu in the COP layer was wet-etched using an etchant (H$_3$PO$_4$ (85%):HNO$_3$ (70%):acetic acid:H$_2$O = 16:1:1:2) to form a Cu channel. An annealing process was then performed under an Ar atmosphere at 300 °C for 2 h. Finally, the laminated film was peeled from the glass substrate to obtain a complete FCF sensor.

### CFD simulation analysis
To analyse the temperature distribution of the proposed sensor and compare it to that of our 1D heat transfer model, we performed a 2D thermofluid coupling CFD simulation. A computational domain with a width of 100 mm and height of 75 mm was used, and a temperature constraint of 300 K was applied to the four walls. The inlet was defined on the left wall of the computational domain to apply the incoming flow. The temperature difference curve was obtained by extracting the temperature at the points on the horizontal centreline of the heater and thermistors. For the simulation of three different configurations (suspended, cavity, and solid structures), the section heights and widths and applied boundary conditions were kept consistent.

To analyse the directionality characteristics of the FCF sensor, a 3D thermofluid coupling CFD simulation was performed. The dimensions of the computational domain were 0.5 × 0.5 × 0.2 m (L × W × H). The sensor and fixture dimensions were kept consistent with those in our experiments, and a temperature constraint of 300 K was applied to the fixture. The temperature differences of V1–3 were calculated by extracting the average temperatures on the centreline of the thermistors at different flow velocities and angles. The physical parameters of the fluid and material required for simulation are listed in Supplementary Table 4. The software we used was ANSYS2020-R2.

### Measurement of the FCF sensor
The performance of the FCF sensor was characterised using a desktop wind tunnel (WT4401-D, OMEGA, USA). The output signal of the FCF sensor was input into a DAQ card (USB-6366, NI) through a differential amplifier circuit, acquired by the host computer, and then subjected to 5 Hz low-pass digital filtering.

The FCF sensor was attached to the leading edge of a wing for AOA and AOS estimation. The cross section of the leading edge of the NACA 0016-MOD airfoil was approximated as a circle with a radius of 14 mm. Airfoil angles relative to the airflow were controlled using two rotation stages.

Two FCF sensors and a commercial IMU (MPU-6050) module were mounted on the MAV to measure the flight parameters. The output signals from the FCF sensors were differentially amplified and digitised (ADS1256). The microcontroller (ESP32-2S) simultaneously acquired the signals from the FCF sensor and digital signals from the IMU module at a sampling frequency of 50 Hz and stored them in an SD card. Ten reflective markers were attached to the fuselage to obtain the 3D coordinates of the MAV. The flight velocity and attitude were calculated using the 3D reconstructed points of the markers obtained from a multi-camera motion capture system (Optitrack, Corvallis, OR, USA).

### MLP neural network algorithm
The MLP used is a fully connected artificial neural network model consisting of an input layer, several hidden layers, and an output layer. Each hidden layer consists of several neuron nodes, and each neuron weights the nodes of the previous layer and performs a nonlinear mapping with an activation function to achieve a multilayer nonlinear regression task from input data to output objects[44,45]. The MLP neural network used was constructed through the Pytorch framework.

For AOA and AOS estimation, a series of MLP neural networks were trained to estimate the AOA and AOS values of the airfoil using data collected from the FCF sensor during wind tunnel testing to evaluate the estimation accuracy for each input pattern (PV1, PV2, PV3, and PV1–3). The inputs of each network were the normalised values of the corresponding signal channels, and the outputs were the AOA and AOS values. To avoid estimation errors caused by different network structures, the same learning rate, activation function, hidden layer, and number of nodes were set for each network[12,46] (see Supplementary Table 5 for details of network structures and training parameters). Relatively small networks were selected to reduce the risk of overfitting.

In the flight velocity estimation experiment, eight channels of data from the P1–2 and V1–2 outputs of two FCF sensors were used as inputs to train the MLP neural network to estimate the flight velocity of the MAV, and the outputs were the three velocity components in the body coordinate system. After training and optimisation, the network model was setup with three hidden layers, and the number of nodes in each layer was 100. The network structures and training parameters are shown in Supplementary Table 5. Training was performed by combining both indoor and outdoor test flight data. All data from the slide reciprocating motion, indoor flight, and outdoor flight tests were combined in the training set to improve the estimation performance of the MLP because the velocity ranges and testing conditions (e.g., propeller spinning and atmospheric conditions) between datasets were different. The estimation performance was validated using indoor and outdoor flight datasets collected independently of those used for training.

### Reporting summary
Further information on research design is available in the Nature Portfolio Reporting Summary linked to this article.

## Data availability

The training and testing datasets of MLP network models generated in this study have been deposited in the figshare database under accession code: https://doi.org/10.6084/m9.figshare.24925971.v1[47]. Source data are provided with this paper.

## Code availability

The codes for training and testing MLP network models have been deposited in Zenodo repository: https://doi.org/10.5281/zenodo.10681822[48].

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

## Acknowledgements

This work was supported financially by the National Key Research and Development Program of China (2023YFB3208000 received by Y.J.), and the National Natural Science Foundation of China (Nos. T2121003, 52022008, 51975030 received by Y.J.).

## Author contributions

Z.G. and Y.J. conceived the project and designed the experiments. Z.G. and Y.J. performed the experiments and analysed the experimental data. Z.G., Z.D. and H.Y. performed the computational fluid dynamics simulation. Z.G. and H.Y. performed thermistor material characterisation and sensor fabrication. Z.G. and Z.Y. carried out wind tunnel experiments. Z.G., W.D., Z.T., H.Z., H.L. and Z.W. designed the MAV flight velocity estimation experiment and analysed the experimental data. Z.G. and Y.J. wrote the manuscript. D.Z., X.D., H.C., D.L. and J.X. proofread the manuscript.

## Competing interests

The authors declare no competing interests.
