## [Peer Review File · Nature Communications]

REVIEWER COMMENTS

Reviewer #1 (Remarks to the Author):

The manuscript entitled “Flexible Calorimetric Flow Sensor with Unprecedented Sensitivity and Directional Resolution for Multiple Flight Parameter Detection” by Gong et al. discusses the development of flow sensor with a heater and equidistant thermistors on flexible substrate for the simultaneous detection of flow speed and direction. The authors, after covering the current theoretical background with a solid introduction, describe the development of the device, from conceptualization to fabrication and evaluation. Additionally, the proposed sensor system is evaluated under true conditions, being installed onto a wing model, for extracting angle of attack (AOA) and angle of slip (AOS), and is finally taken for a flight using a real MAV. The data is always checked against commercial sensors. The authors report detection of velocity with a resolution of 0.11 mm/s and an angular resolution of .0.1o when estimating flow direction.

This work uses patterned microchannels under the sensor structure, allowing for a large temperature difference and very efficient power requirement for the heater, having so thin thermal mass under it. As it can be seen from the relevant, comparing calorimetric sensors, there have been numerous approaches for micromachining such structures; thus this element might be, in my humble opinion, a very interesting micromechanical feat, but is not at all a new concept. For further rendering the sensor performance, the designers are left with the following major design parameters: a) heater and sensor materials (here VOx), sensor placement with respect to heater (here well-discussed dimension “D”), substrate physical dimensions and driving mode for the heater. Let me argue on the aforementioned, and if the manuscript adds new knowledge. The following apply to “Results” first paragraphs.

For material selection, I feel that the authors have not adequately argued with regard to selecting VOx. The only relevant discussion in the manuscript is “VOx was employed as the material for integrated thermistors on a flexible substrate because it exhibits a high TCR and low $1/f$ noise, providing significant advantages for temperature sensing applications”. I encourage the authors to elaborate more on this, there are numerous other approaches in material selection with higher TCR. How was the thermistor number decided? Why did the authors choose three thermistors per side and not, say, ten? I understand that this can easily be interpreted as a trade-off between readout system, optimal performance and measurement range, but the design decisions need equal attention. The thermistor positioning is well-discussed and backed with theory, but is not something new. Finally, the driving electronics and the mode of operation are state-of-the art, and can be found in other similar works as well.

Now for comments on the experiment series: for the first series of experiments, it is my opinion that breathing over the sensor is not a repeatable experiment. Was the breathing experiment conducted under open environment? The flow is not laminar, there is a lot of humidity in the fluid passing over the device, the angle of attack is not constant, the flow is different for each breath and each person. Please add some comments if you intend to keep this.

For the second experiment, it is very well presented, and a nice experimental way that couples with the theoretical work for extracting the functions for each D distance from the heater.

For AOS and AOA experiments, please comment on the actual feasibility of running this algorithm on software that can accompany the sensor. How much time is needed for inference? Is that time enough for using the result as part of a decision support system?

Finally, for the installed on MAV experiment, it is an impressive amount of work and leaves no doubt that the proposed system is capable of extracting very small fluctuations in air speed. Just some information for declaration, excuse me if I missed it: where were these calculations performed? On-board or offline? What was the process for actually extracting the end data presented, such as the vibrating frequency etc.? Also, discussion needs to be added about the limits of the system with respect to losses (heat loss coefficient etc.).

Discussion refers to structural optimization, which, in my opinion should normally include iterations and measurements of devices. Other than that, the points raised here are all valid.

The work presented in this manuscript is in my opinion of high quality. It presents approaches that are known to the scientific community and have already been presented in some form. The authors themselves provide a good list of references for similar work. I think that the authors locate their innovation in using the MLP and the actual proof of work with installing the system onto the MAV. The flow of the manuscript is fine, the rationale exists and is backed with theoretical approach and design, followed by experimental data. The methodology is sound and all the relevant information is available so as to be reproduced by anyone. There is no doubt that the work is of very high quality, I have some doubts regarding the importance and innovation (solely by keeping in mind that such systems have been presented again). Nevertheless, I propose publication after these points have been discussed because such flexible and integrated system has never been manufactured and characterized in the rigorous manner presented herein.

A list with some comments follows:

Figure 1: A) I would prefer a simpler representation; the actual microstructure of the sensor is lost in the bigger scale of the device. Maybe a detailed view in-set would be beneficial

B) Is there a reason not to present a top-view? In my opinion the message would get across in a better manner.

C) Legend "D" is, to me, very unclear. I struggled finding it.

D) This looks like a rendering. Also, where are the readout wires?

Figure 2: A) The coloring of the thermistors needs to be a lot bolder.

Supplementary Figure 1. A. Please, zoom in. This figure is exactly what I was missing as a reader in the manuscript.

Supplementary Figure 3. It would be very helpful to add profile (or front) views from each step. Helps to grasp the inner layout of the sensor.

Supplementary Table 2, 7th work has a reference of "00".

What software was used for the CFD?

How was the data recorded on-board? What was the sampling rate for (both the commercial and the custom) sensors?

What tool was used for training and testing the neural network?

Wishing you the best of luck and congratulations on your work.

Reviewer #2 (Remarks to the Author):

The authors present a novel sensor for estimating the velocity and attitude of UAVs based on airflow. The estimation of these parameters is crucial for autonomous and semi-autonomous flight. Existing solutions are often based on a combination of vision and MEMS-based IMUs, whereas small and cheap IMUs are very noisy, and the use of vision requires a camera setup and a significant amount of compute power. Thus, lightweight alternatives are an important contribution to the miniaturization of UAVs.

After a brief problem-statement follows a detailed discussion on existing solutions to sensing flight parameters. The challenges of existing solutions are explained and details on the expected performance are given as supplementary material. An experimental comparison with an off-the-shelf sensor is performed, showing the advantages of the proposed solution. The main contribution seems to be the increased accuracy especially in slow-flight situations. Details on the manufacturing process of the sensor and on the sensor itself are given in the article and extended in the supplementary material, as well as parameters for modeling the simulation and the parametrization of the sensor.

A pre-trained machine learning model is used to calculate the attitude based on current measurements. Here, a brief description on what model is used is lacking in the methodology, as well as an in-depth explanation and references to the used model. I would like an elaboration on these aspects and how the approach has been tailored to the given problem.

The evaluation is performed in simulation, in a wind tunnel for the sensor itself, and inflight experiments for the overall system performance. The results are backing the authors' claims and show that the proposed sensor system achieves a good performance. Real-flight experiments are documented in two supplementary videos. The videos give a good impression on the uses UAV, nevertheless from the text alone the flight characteristics were not clear to me. I would suggest to briefly describe what type of MAV is considered (e.g., rotary, fixed-wing, combination) in the introduction.

To assess the drift, I'd appreciate an integration of the flight path over time, compared to the ground-truth or GPS position of the MAV.

The article is well-written and understandable.

Thanks for your comments concerning our manuscript entitled “**Flexible Calorimetric Flow Sensor with Unprecedented Sensitivity and Directional Resolution for Multiple Flight Parameter Detection**” (No. NCOMMS-23-43845). Those comments are valuable for revising and improving our paper with important guiding significance. We have made correction according to the comments, revised portion are marked in **red** in the manuscript. We list your valuable comments with our responses shown in **blue**.

The authors present a novel sensor for estimating the velocity and attitude of UAVs based on airflow. The estimation of these parameters is crucial for autonomous and semi-autonomous flight. Existing solutions are often based on a combination of vision and MEMS-based IMUs, whereas small and cheap IMUs are very noisy, and the use of vision requires a camera setup and a significant amount of compute power. Thus, lightweight alternatives are an important contribution to the miniaturization of UAVs.

After a brief problem-statement follows a detailed discussion on existing solutions to sensing flight parameters. The challenges of existing solutions are explained and details on the expected performance are given as supplementary material. An experimental comparison with an off-the-shelf sensor is performed, showing the advantages of the proposed solution. The main contribution seems to be the increased accuracy especially in slow-flight situations. Details on the manufacturing process of the sensor and on the sensor itself are given in the article and extended in the supplementary material, as well as parameters for modeling the simulation and the parametrization of the sensor.

Response: We greatly appreciate the time and valuable comments of the reviewer. We have revised our manuscript based on your comments and hope that the updated manuscript addresses your comments.

A pre-trained machine learning model is used to calculate the attitude based on current measurements. Here, a brief description on what model is used is lacking in the methodology, as well as an in-depth explanation and references to the used model. I would like an elaboration on these aspects and how the approach has been tailored to the given problem.

Response: Thank you very much for the comment. We have further elaborated and explained the principles and building of the MLP neural network model in the Methods on page 21 of the

manuscript. We are convinced that parameters such as activation function, number of hidden layers and nodes can influence the estimation results of the algorithm. Therefore, to evaluate the effect of the number of thermistor pairs on the estimation accuracy of AOA and AOS in a more reasonable way, we used a series of neural network models with the same structure and training parameters (Supplementary Table 4) to train and test the estimation results for different input patterns (see page 13 of the manuscript for details). The results remain as expected, i.e., the use of thermistor arrays is beneficial for improving the estimation accuracy of AOA and AOS.

The evaluation is performed in simulation, in a wind tunnel for the sensor itself, and inflight experiments for the overall system performance. The results are backing the authors' claims and show that the proposed sensor system achieves a good performance. Real-flight experiments are documented in two supplementary videos. The videos give a good impression on the uses UAV, nevertheless from the text alone the flight characteristics were not clear to me. I would suggest to briefly describe what type of MAV is considered (e.g., rotary, fixed-wing, combination) in the introduction.

Response: Thank you very much for the comment. The MAV presented in this article is similar to a tail-sitter aircraft. We prefer to refer to it as a belly-sitter. It has two vector propellers to meet all the control and propulsion needed. When landing, it lowers vertically like a quad-rotor, then gently touches the ground, flips and lands on its belly. During takeoff, the drone can be pulled up from the ground by deflecting the vector mechanism perpendicular to the fuselage. According to the other drones name conventions, it is more appropriate to call it as fly-wing MAV with dual vector rotors. On page 15 of the manuscript, we added a description of the type of MAV.

To assess the drift, I'd appreciate an integration of the flight path over time, compared to the ground-truth or GPS position of the MAV.

Response: Thank you very much for the comment. First, to further reduce the estimation error of the relative airflow velocity, we optimized the neural network structure. After training and testing, the number of nodes per hidden layer was increased from 50 to 100. As shown in **Fig. R1**, the estimation error of the flight velocity components of each axis were significantly decreased based on the optimized neural network. We updated the optimized estimates to Fig. 5 of the manuscript.

Fig. R1 MAE of the indoor and outdoor flight velocities between the actual and estimated values. (A) Previous network model 1 with 3 hidden layers and 50 nodes per layer; (B) Optimized network model 2 with 3 hidden layers and 100 nodes per layer.

Then, to evaluate indoor and outdoor flight path drift, we performed integration of the velocity estimates. Since these estimates are given in body coordinate frame, they were first transformed in inertial frame using the rotation matrix, and then numerically integrated.

Fig. R2 Flight path estimation and drift assessment. (A) Drift of path estimation obtained by flight velocity (Airflow) with respect to the ground truth given by motion capture during indoor flight phases. (B) Drift of the path estimation obtained by velocity integration with respect to the path based on EKF and GNSS during outdoor flight. (C) Path obtained by velocity integration versus the path given by EKF and GNSS during outdoor flight.

Fig. R2A displays the drift of the path obtained by flight velocity with respect to the ground truth during indoor flight phases. The mean drifts of Euclidean distance to the ground truth for the three flight phases were calculated to be 0.0692, 0.0496, and 0.0639 $\text{m}\cdot\text{s}^{-1}$, respectively. This reflects the feasibility to use the obtained flight velocity estimates for odometry. **Fig. R2B** shows the drift of the path estimation obtained by velocity integration with respect to the path based on EKF and Global Navigation Satellite System (GNSS) during outdoor flight. **Fig. R2C** shows the path obtained by velocity integration versus the path given by EKF and GNSS during outdoor flight. The average drift of the path based on velocity integration to the path given by EKF and GNSS were calculated to be 0.1161 and 0.4872 $\text{m}\cdot\text{s}^{-1}$, respectively. The MLP network for flight velocity estimation is pre-trained with the velocity given by the EKF, therefore the path estimation based on flight velocity is closer to the path given by the EKF. However, due to the unavoidable drift of consumer-grade GNSS, IMUs, etc. ^{1,2}, it is difficult for us to assess the drift between the estimated path and the ground truth for outdoor flight. In summary, the indoor path estimation results reflect the feasibility of using the obtained flight velocity estimates for odometry. In future work, we will consider how to optimize the path estimation algorithm for fused flight velocity to further improve the practicality of FCF sensors for flight parameter detection.

Thank you again for your valuable comments on the FCF sensor, which makes this work better accessible to *Nat. Commun.* readers.

Reference

1. Wood, K. T., Araujo-Estrada, S., Richardson, T. & Windsor, S. Distributed Pressure Sensing–Based Flight Control for Small Fixed-Wing Unmanned Aerial Systems. *Journal of Aircraft* **56**, 1951–1960 (2019).
2. Borup, K. T., Fossen, T. I. & Johansen, T. A. A Machine Learning Approach for Estimating Air Data Parameters of Small Fixed-Wing UAVs Using Distributed Pressure Sensors. *IEEE Trans. Aerosp. Electron. Syst.* **56**, 2157–2173 (2020).

Thanks for your comments concerning our manuscript entitled “**Flexible Calorimetric Flow Sensor with Unprecedented Sensitivity and Directional Resolution for Multiple Flight Parameter Detection**” (No. NCOMMS-23-43845). Those comments are valuable for revising and improving our paper with important guiding significance. We have made correction according to the comments, revised portion are marked in red in the manuscript. We list your valuable comments with our responses shown in blue.

The manuscript entitled “Flexible Calorimetric Flow Sensor with Unprecedented Sensitivity and Directional Resolution for Multiple Flight Parameter Detection” by Gong et al. discusses the development of flow sensor with a heater and equidistant thermistors on flexible substrate for the simultaneous detection of flow speed and direction. The authors, after covering the current theoretical background with a solid introduction, describe the development of the device, from conceptualization to fabrication and evaluation. Additionally, the proposed sensor system is evaluated under true conditions, being installed onto a wing model, for extracting angle of attack (AOA) and angle of slip (AOS), and is finally taken for a flight using a real MAV. The data is always checked against commercial sensors. The authors report detection of velocity with a resolution of 0.11 mm/s and an angular resolution of 0.1° when estimating flow direction.

This work uses patterned microchannels under the sensor structure, allowing for a large temperature difference and very efficient power requirement for the heater, having so thin thermal mass under it. As it can be seen from the relevant, comparing calorimetric sensors, there have been numerous approaches for micromachining such structures; thus this element might be, in my humble opinion, a very interesting micromechanical feat, but is not at all a new concept. For further rendering the sensor performance, the designers are left with the following major design parameters: a) heater and sensor materials (here VO_x), sensor placement with respect to heater (here well-discussed dimension “D”), substrate physical dimensions and driving mode for the heater. Let me argue on the aforementioned, and if the manuscript adds new knowledge. The following apply to “Results” first paragraphs.

Response: We greatly appreciate the time and valuable comment of the reviewer We have revised our manuscript based on your comments and hope that the updated manuscript addresses your comments.

For material selection, I feel that the authors have not adequately argued with regard to selecting VO_x. The only relevant discussion in the manuscript is “VO_x was employed as the material for integrated thermistors on a flexible substrate because it exhibits a high TCR and low 1 / f noise, providing significant advantages for temperature sensing applications”. I encourage the authors to elaborate more on this, there are numerous other approaches in material selection with higher TCR.

Response: Thank you very much for the comment. We list the TCR and resistivity properties of some thermal sensing materials in Supplementary Table 2. High TCR thermistor materials such as amorphous silicon-germaniums can indeed significantly improve the sensitivity of FCF sensors, but 1 / f noise is also of concern. Usually, 1 / f noise is positively correlated with material resistance, so we chose VO_x, which combines high TCR and low resistivity, as the thermistor material. On page 6 of the manuscript, we further elaborate on the basis for the selection of thermistor materials. In addition, there are studies demonstrating that VO₂(B) can achieve higher TCR ($-7\% \text{ K}^{-1}$) and lower resistivity ($<0.01 \text{ } \Omega \cdot \text{m}$), which are more suitable to be used as a thermosensitive sensing material^{1,2}. This also provides a direction for our future research.

How was the thermistor number decided? Why did the authors choose three thermistors per side and not, say, ten? I understand that this can easily be interpreted as a trade-off between readout system, optimal performance and measurement range, but the design decisions need equal attention. The thermistor positioning is well-discussed and backed with theory, but is not something new. Finally, the driving electronics and the mode of operation are state-of-the art, and can be found in other similar works as well.

Response: Thank you very much for the comment. In the sensor design stage, the number of thermistors was decided based on our consideration of the velocity measurement range and sensor response characteristics. Firstly, the results of simulations and calculations show that the optimal distance D between the heater and thermistor pair with the maximum temperature difference ΔT_{max} decreases as the flow velocity increases. When the incoming flow is $0.02 \text{ m} \cdot \text{s}^{-1}$, the distance D with ΔT_{max} is about $500 \text{ } \mu\text{m}$, so we consider placing the thermistor within this distance to cope with the tiny flow velocity measurement during MAV flight. Secondly, to obtain non-linearly correlated data for more accurate angle estimation, different thermistor pairs are required to have a significant difference in the response characteristics. This requires that neighboring thermistors be spaced as far apart as possible. Finally, considering the width of the electrode wires and process stability, we

retained three pairs of thermistors with distances D from the heater of 120, 320 and 520 μm , respectively. In addition, we marked the positions of three thermistor pairs at the corresponding D in Fig. 2A and Supplementary Fig. 1E.

Now for comments on the experiment series: for the first series of experiments, it is my opinion that breathing over the sensor is not a repeatable experiment. Was the breathing experiment conducted under open environment? The flow is not laminar, there is a lot of humidity in the fluid passing over the device, the angle of attack is not constant, the flow is different for each breath and each person. Please add some comments if you intend to keep this.

Response: Thank you very much for the comment. The breathing airflow test reflects the small airflow sensing capability of the FCF sensor, but its test object, conditions, etc. are not consistent with the subject of relative airflow and flight parameter measurements for MAV. After careful consideration we decided to delete the breathing airflow test experiment to make the article more coherent and consistent.

For the second experiment, it is very well presented, and a nice experimental way that couples with the theoretical work for extracting the functions for each D distance from the heater.

Response: Thank you very much for the encouraging remarks.

For AOS and AOA experiments, please comment on the actual feasibility of running this algorithm on software that can accompany the sensor. How much time is needed for inference? Is that time enough for using the result as part of a decision support system?

Response: Thank you very much for the comment. The estimation of AOA and AOS was performed on a computer. The inference time of the artificial neural network algorithms on a computer does not seem to reflect their actual feasibility because the performance of the onboard processor of the MAV is hardly comparable to that of a computer. To assess the algorithmic feasibility of AOA and AOS estimation, we ported two trained neural network models to the microprocessor (STM32F405RGT6) used by the MAV and simulated the real-time inference process, and the results are shown in Table R1. It can be shown that the network consisting of 3 hidden layers with 50 nodes per layer has a single processing time of 0.318 ms. After increasing each hidden layer to 100 nodes, the single processing time is 1.065ms. Based on above results, we

consider it feasible to use the inference results for the decision support system for MAV.

Table R1 Comparison of the performance of existing flight parameter detection studies.

Parameters	Model for AOA and AOS estimation	Model for relative airflow velocity estimation
Number of input nodes	6	8
Number of hidden layers and nodes	3×50	3×100
Number of output nodes	2 (AOA and AOS)	3 (Velocity x, y and z)
Activation function	ReLU	ReLU
Number of samples	10000	10000
Total processing time	3.18 s	10.65 s
Single processing time	0.318 ms	1.065 ms

Finally, for the installed on MAV experiment, it is an impressive amount of work and leaves no doubt that the proposed system is capable of extracting very small fluctuations in air speed. Just some information for declaration, excuse me if I missed it: where were these calculations performed? On-board or offline? What was the process for actually extracting the end data presented, such as the vibrating frequency etc.? Also, discussion needs to be added about the limits of the system with respect to losses (heat loss coefficient etc.).

Response: Thank you for the comment. The actual velocity of indoor flight was calculated offline from the MAV flight trajectory recorded by the motion capture system after differentiation and coordinate transformation. The actual velocity of outdoor flight was calculated offline by EKF algorithm based on the information recorded by onboard satellite positioning system, IMU, and manometer. The estimation of the MAV flight velocity was performed offline based on the pre-trained MLP neural network.

The process of wing vibration measurement can be described as follows: when the MAV is hovering, the on-board ESP32-2S simultaneously acquires the three-axis acceleration information from the two FCF sensors and the MCU at a sampling frequency of 50Hz and stores it in the SD card. After acquiring the data offline, we processed the FCF sensor signals with 2Hz high-pass

digital filtering. Fig. 5F shows the output voltage of P1 (sensitive to z-axis velocity component) of FCF sensor #2, the 2 Hz high-pass filtered signal, and the acceleration signal acquired by the IMU synchronously during MAV flight. Finally, the short-time Fourier transforms of the filtered sensor signal and MCU acceleration signal were computed to obtain Fig. 5G. We added the IMU acceleration signal in Fig. 5F and declared on page 18 of the manuscript that the short-time Fourier transform method was used to obtain the spectrogram.

Fig. R1 Heating power in the flow velocity range of 0–30 m·s⁻¹.

Regarding system losses, we evaluated the heating power of the FCF sensor at different flow velocities, as shown in **Fig. R1**. It can be seen that the heating power increases from 3 to 4.4 mW in the flow velocity range of 0–30 m·s⁻¹. And the power consumption of this heater can be fitted by King's law at constant heating temperature difference $T_h = 20$ K. Supplementary Table 3 lists the heating power consumption of existing calorimetric flow sensors, where our FCF sensor has the lowest power consumption of the existing 2D flow sensors. Low consumption is also very important for the energy conservation and flight endurance of the MAV. We have added a description of heat loss on page 9 of the manuscript.

Discussion refers to structural optimization, which, in my opinion should normally include iterations and measurements of devices. Other than that, the points raised here are all valid.

Response: Thank you for the comments. We have added a description related to integrated measurement systems in the discussion section.

The work presented in this manuscript is in my opinion of high quality. It presents approaches that are known to the scientific community and have already been presented in some form. The authors themselves provide a good list of references for similar work. I think that the authors locate their innovation in using the MLP and the actual proof of work with installing the system onto the

MAV. The flow of the manuscript is fine, the rationale exists and is backed with theoretical approach and design, followed by experimental data. The methodology is sound and all the relevant information is available so as to be reproduced by anyone. There is no doubt that the work is of very high quality, I have some doubts regarding the importance and innovation (solely by keeping in mind that such systems have been presented again). Nevertheless, I propose publication after these points have been discussed because such flexible and integrated system has never been manufactured and characterized in the rigorous manner presented herein.

Response: Thank you for the insightful remarks.

A list with some comments follows:

Figure 1: A) I would prefer a simpler representation; the actual microstructure of the sensor is lost in the bigger scale of the device. Maybe a detailed view in-set would be beneficial.

B) Is there a reason not to present a top-view? In my opinion the message would get across in a better manner.

C) Legend “D” is, to me, very unclear. I struggled finding it.

D) This looks like a rendering. Also, where are the readout wires?

Figure 2: A) The coloring of the thermistors needs to be a lot bolder.

Supplementary Figure 1. A. Please, zoom in. This figure is exactly what I was missing as a reader in the manuscript.

Supplementary Figure 3. It would be very helpful to add profile (or front) views from each step. Helps to grasp the inner layout of the sensor.

Supplementary Table 2, 7th work has a reference of “00”.

What software was used for the CFD?

How was the data recorded on-board? What was the sampling rate for (both the commercial and the custom) sensors?

What tool was used for training and testing the neural network?

Response: Thank you very much for the comments.

For Fig. 1: A) we added a zoomed-in view of the heater and thermistor array structure.

B) We show a top-view of the simulation to better represent the flow directionality detection.

C) We have relabeled Legend “D” to make it look clearer.

D) We retook the photo of the FCF sensor attached to the glass rod with the wires exposed.

For Fig. 2: A) We have labeled the VO_x thermistors in a more vibrant color.

For Supplementary Fig. 1A: We show a zoomed-in view of the FCF sensor structure.

For Supplementary Fig. 3: We added corresponding profile views for each process step.

For Supplementary Table 2: We have corrected the serial numbers of the references.

CFD simulations were performed using ANSYS2020-R2. We have made an addition on page 20 of the manuscript.

The microcontroller acquires data from FCF sensors and commercial IMU module synchronously at a sampling frequency of 50 Hz and stores the data on an on-board SD card. We have made an addition on page 21 of the manuscript.

We trained and tested the MLP neural network using the Pytorch framework, as declared on page 21 of the manuscript.

Thank you again for your valuable comments on the FCF sensor, which makes this work better accessible to *Nat. Commun.* readers.

Reference

1. Chen, S. et al. Characterization of nanostructured VO₂ thin films grown by magnetron controlled sputtering deposition and post annealing method. *Opt. Express* **17**, 24153 (2009).
2. Zhang, Y., Xiong, W., Chen, W. & Zheng, Y. Recent Progress on Vanadium Dioxide Nanostructures and Devices: Fabrication, Properties, Applications and Perspectives. *Nanomaterials* **11**, 338 (2021).

REVIEWERS' COMMENTS

Reviewer #1 (Remarks to the Author):

After carefully going through the manuscript with the authors response, I am confident that it suits the quality of the journal very well.

I can see that all the comments have been addressed effectively and the authors gave the appropriate attention to detail.

Therefore I propose to move forward with publishing the article.

Reviewer #2 (Remarks to the Author):

The authors revised their manuscript and addressed my open questions adequately. They extended the article with useful information and provided additional supplementary material with parameters and results.

If possible I would encourage the authors to add the information given in Fig. R2 in their response letter to the article or supplemental material.

Thanks for your comments concerning our manuscript entitled “**Flexible Calorimetric Flow Sensor with Unprecedented Sensitivity and Directional Resolution for Multiple Flight Parameter Detection**” (No. NCOMMS-23-43845). Those comments are valuable for revising and improving our paper with important guiding significance.

After carefully going through the manuscript with the authors response, I am confident that it suits the quality of the journal very well.

I can see that all the comments have been addressed effectively and the authors gave the appropriate attention to detail.

Therefore I propose to move forward with publishing the article.

Response: We greatly appreciate the time and valuable comment of the reviewer, which makes this work better accessible to *Nat. Commun.* readers.

Thanks for your comments concerning our manuscript entitled “**Flexible Calorimetric Flow Sensor with Unprecedented Sensitivity and Directional Resolution for Multiple Flight Parameter Detection**” (No. NCOMMS-23-43845). Those comments are valuable for revising and improving our paper with important guiding significance.

The authors revised their manuscript and addressed my open questions adequately. They extended the article with useful information and provided additional supplementary material with parameters and results.

If possible I would encourage the authors to add the information given in Fig. R2 in their response letter to the article or supplemental material.

Response: We greatly appreciate the time and valuable comments of the reviewer. We have incorporated the contents of Fig. R2 into the main text and Supplementary Figure 9 in accordance with your suggestion.

Thank you again for your valuable comments, which makes this work better accessible to *Nat. Commun.* readers.